# The Impact of Knee Orthoses on Lameness and Weight Distribution in Canine After Rupture of the Cranial Cruciate Ligament

**DOI:** 10.3390/ani15040545

**Published:** 2025-02-13

**Authors:** Aljaž Muršec, Monika Pavlović, Tomaž Lampe, Vladimira Erjavec

**Affiliations:** 1Faculty of Health Sciences, University of Ljubljana, 1000 Ljubljana, Slovenia; aljaz.mursec@zf.uni-lj.si (A.M.); tomaz.lampe@zf.uni-lj.si (T.L.); 2Department of Prosthetics and Orthotics, Metropolia University of Applied Sciences, 00920 Helsinki, Finland; monika.pavlovic4@gmail.com; 3Veterinary Faculty, University of Ljubljana, 1000 Ljubljana, Slovenia

**Keywords:** cranial cruciate ligament rupture, orthotics, dog, veterinary orthopedics, canine rehabilitation

## Abstract

Rupture of the cranial cruciate ligament is a common orthopedic problem in dogs that severely affects joint stability and quality of life. This study investigated the use of custom-made knee orthoses as a conservative treatment for recent injuries. Two dogs were fitted with custom orthoses, and their recovery was monitored over a 32-day period using weight distribution measurements and owner assessments of mobility, gait, and lameness. Both dogs showed significant improvement in limb function, with one of them achieving near symmetrical weight bearing at the end of the study. Shifts in the weight distribution of the front and hindlimbs showed better lateral asymmetry, further confirming progress. Owner feedback was consistent with these results, indicating noticeable improvements in mobility and reduced lameness. The results show that knee orthoses have the potential to be an effective non-surgical alternative for the treatment of ligament injuries and may improve the quality of life of dogs. This study highlights the need for further research to optimize orthotic solutions in veterinary medicine.

## 1. Introduction

Rupture of the cranial cruciate ligament (CCL) is among the most common orthopedic conditions in veterinary medicine [1]. This ligament plays an important role in maintaining the stability and biomechanics of the stifle joint, and its abnormal function significantly affects the animal’s quality of life [2]. In dogs, the weakening of the stifle joint structure resulting from the progressive degeneration of the ligament is the most commonly recognized etiopathogenetic hypothesis, which is related to factors such as breed, sex, age, and the shape of the stifle joint [3]. Treatment options for CCL rupture can be conservative or surgical, with the choice depending on several factors, including the dog’s age, overall health, degree of joint instability, and financial considerations [4]. Conservative methods, such as physiotherapy, play an important role in rehabilitation by focusing on pain relief, restoring normal range of motion, strengthening periarticular and stabilizing muscles, and addressing proprioceptive disorders. These goals are similar across species, including humans and dogs [5]. Recently, knee orthoses have gained popularity as a non-surgical alternative for managing knee ligament injuries, offering promising results for both rehabilitation and stabilization [6].

Orthoses are increasingly used in veterinary medicine for various orthopedic conditions, depending on the specific needs of the patient and the nature of the injury or disease [7]. Their primary functions include restricting, controlling, or supporting limb movement, thereby stabilizing joints, relieving pain, and preventing further injury. Additionally, orthoses act as a protective shield, reducing stress on damaged or vulnerable tissues and shielding them from external factors [8]. In veterinary clinical practice, orthoses are often used to stabilize joints affected by conditions such as a ruptured CCL, hip or knee dysplasia, ligament injuries, and fractures [9]. They also play a significant role in post-surgical rehabilitation, enabling controlled movements that facilitate healing while minimizing the risk of re-injury [10]. Beyond their therapeutic application, orthoses are invaluable in preventive care. They help prevent joint overload, reduce the risk of further injuries, and improve overall limb functionality by maintaining proper biomechanics during movement [11]. However, knee orthoses can also cause some side effects, such as skin irritation, the limited mobility of adjacent joints, or the inability of the animal to accept mobility aids, which is less common than other effects [12,13,14].

The aim of this study is to evaluate the potential effectiveness of knee orthoses in the rehabilitation of recent CCL rupture in dogs. It specifically examines their impact on body weight distribution across the limbs over a one-month period. A combination of the four-scale method and an owner questionnaire was utilized to gather data.

## 2. Materials and Methods

Before the orthoses were made and measurements were performed, the owners of the dogs signed a consent form, agreeing to the use of the orthoses and participation in this study.

The study included two dogs of similar size and body weight and with the same injury (rupture of the CCL), which occurred around the same time. The dogs underwent X-rays and a manual laxity test performed by veterinarians from two private veterinary clinics to confirm the diagnosis. The physical characteristics, details of the injuries, and therapies are summarized in Table 1.

### 2.1. Measurements and Production of Knee Orthoses

To stabilize the knee joint, two custom-made orthoses were created, tailored precisely to fit the shape of each dog’s limbs (Figure 1).

Before the orthoses were manufactured, precise measurements of the injured limb were taken, including the limb’s length (from the hip to the knee joint, from the knee joint to the heel, and from the heel to the toes) and the circumference of the limb at 2 cm intervals. A negative model of the limb was taken using plaster bandages, which was then filled with plaster to produce a positive model. The plaster positive was processed in such a way that all bone points were relieved, thus avoiding damage to the skin. Polyform foam 4 mm was applied to the model, 3D-printed joints were added in the knee joint area (medial and lateral), and polypropylene plastic 4 mm was applied over the model using the heating method (170 °C) and vacuum. The polypropylene was cut out of the model together with the foam and the joints, the edges were sanded, and 4 Velcro fasteners were attached (3 on the posterior side, 1 on the anterior side at the level of the patellar ligament). The orthoses were designed with a four-point force system to provide anteroposterior stabilization of the stifle joint (Figure 2).

On receipt of the orthosis, owners were provided with instructions on the correct fitting of the orthosis, cleaning, and acclimating their pet to the orthosis. They were advised to use the orthosis only during walks and in a controlled environment for the first few days. Walks were to be adjusted based on the animal’s condition. During the first week, the orthosis was recommended for 10–15 min per walk, with the duration gradually increasing by 10 min if no complications arose. Additionally, owners were instructed to examine the limbs before and after applying the orthosis to check for any skin lesions.

### 2.2. Research Methods

In order to check the condition of each dog, a questionnaire was designed for the owners. The questionnaire collected data on dogs’ lameness, activity levels, and body weight distribution, using a modified assessment method based on Nganvongpanit et al. (2013) [15] (Appendix A). The questionnaire was completed at three time points: the first day of orthosis application, the ninth day at the follow-up examination, and the thirty-second day during the repeat examination. Measurements with scales were also taken at the same intervals.

The transfer of body weight to each limb was measured using four KERN scales (manufacturer: KERN & Sohn GmbH, Balingen, Germany). Each scale measured 305 × 315 mm with a maximum weight capacity of 60 kg and a resolution of 20 g. The scales were covered with non-slip rubber mats and placed on a flat, stable surface. During each measurement session, the dog stepped on the scales six times: first, three times without orthosis, and then, three times with the orthosis (Figure 3). A 3 min rest period was observed between each measurement. The distance between the scales was determined by measuring the dog in a standing position—the length between the front and rear limbs and the length between both front and rear limbs. The dog’s posture was carefully monitored to ensure that all four limbs were placed correctly on the corresponding scale, with the head straight forward and at the same height between each measurement (Figure 3). If the head was lower or higher or rotated to the left or right, there would be an incorrect distribution of weight on the limbs.

Microsoft Excel 2016 software (Microsoft, Redmond, WA, USA) was used to process the data. The average body weight transferred to each limb was calculated from three measurements without orthosis and measurements with orthosis at the respective time points (day 1, day 9, and day 32 of orthosis use). The proportions of body weight distribution across the limbs were calculated using cross-tabulation, providing orthosis’s effectiveness in dogs with CCL ruptures.

## 3. Results

The questionnaire results revealed improvements in all categories, including lameness, gait, mobility, and activity levels, for both dogs on the 32nd day of orthosis use. These results are summarized in Table 2, and a detailed description of lameness assessments is provided in Appendix A.

The body weight distribution measurements using the four-scale method showed improvement in weight-bearing capacity on the injured limb when using the orthosis. By day 32, Dog 1 transferred 0.97 kg more weight to the injured limb than on day 1, and Dog 2 transferred 2.23 kg more. Figure 4 shows dog 1’s weight transfer to the injured and healthy hindlimbs with and without the orthosis over time. Similarly, Figure 5 illustrates Dog 2’s data, showing near-equal weight transfer on the hindlimbs by day 32.

The difference amounted to 0.04 kg. For Dog 1 (Figure 6), on day 1 of orthosis use, 70% of the body weight was transferred to the forelimbs and 30% to the hindlimbs. By day 32, this shifted to 62% on the forelimbs and 38% on the hindlimbs. The data also show that, on average, 62% of the dog’s body weight was borne on the uninjured side (right forelimb and hindlimb) and 38% on the injured side (left forelimb and hindlimb).

For Dog 2 (Figure 7), 60% of the body weight was borne on the forelimbs and 40% on the hindlimbs on day 1, with a shift to 55% on the forelimbs and 45% on the hindlimbs by day 32. On day 1, 59% of body weight was transferred to the contralateral (uninjured) side and 41% to the injured side. By day 32, the distribution was 52% on the uninjured side and 48% on the injured side.

## 4. Discussion

This study investigated whether knee orthoses in dogs aid the rehabilitation of fresh CCL ruptures and how they affect the distribution of body weight across the limbs over a one-month period. Data were collected using a four-scale method and a questionnaire for the owners.

The questionnaire results indicated notable improvements in lameness, gait, and overall activity levels after 32 days of orthosis use. Both dogs demonstrated the ability to walk longer distances and showed better weight distribution across their limbs. These findings align with the normal body weight distribution in dogs, where 60–70% of weight is carried on the forelimbs and 30–40% on the hindlimbs [10,16]. In this study, Dog 1 carried 70% of its body weight on the forelimbs on day 1 and 62% on day 32, while Dog 2 carried 60% of its body weight on the forelimbs on day 1 and 55% on day 32.

Healthy dogs typically distribute their weight evenly between the left and right sides [17]. However, both dogs in this study initially carried more weight on the uninjured side due to the CCL rupture. On day 1, Dog 1 carried 62% and Dog 2 carried 59% of their body weight on the uninjured side. By day 32, Dog 2 showed improvement, bearing 52% on the uninjured side, while Dog 1 exhibited minimal changes. Great attention was paid to the posture of the dogs during all measurements. The height and position of the head (straight forward) were carefully monitored to ensure that there was no incorrect weight shift.

The findings are consistent with previous studies. Adamiak et al. (2022) [18] highlighted high owner satisfaction with knee orthoses, citing increased joint range of motion and the slowed progression of degenerative joint disease. Similarly, Bertocci et al. (2017) [19] reported that knee orthoses stabilized the stifle joint, reducing ligament stress and tibial rotation. Carr et al. (2016) [6] found that knee orthoses are a viable non-surgical alternative, particularly for dogs unable to undergo surgery due to health or financial constraints.

While both dogs showed improvements, Dog 2 achieved better results in weight transfer to the limbs. This may be attributed to the additional conservative therapies (therapeutic massage and laser therapy) that Dog 2 began receiving one week after the injury and throughout the time between measurements, while Dog 1 did not receive such therapies. In a study by Wucherer et al. (2013) [20], they found that conservative methods of treating a CCL rupture (physical therapy, weight loss, and NSAID administration) improved the health of the animals, but surgical methods achieved better results.

Both dogs also received oral joint-strengthening supplements, such as Antinol, after their injury. This suggests that combining orthotic use with complementary rehabilitation methods and oral supplements may enhance recovery outcomes.

Several limitations in this study should be acknowledged. The first is the small sample size: only two dogs were included, limiting the generalizability of the results. The follow-up period was short: the study covered only one month, which may not fully capture the long-term effects of orthosis use. Questionnaire responses might be influenced by owner perception rather than objective measures. The study relied on owner-reported activity levels, which might also not be entirely accurate. There was also variability in therapies: Dog 2 received additional therapies, which could confound the results, making it difficult to isolate the effects of the orthosis. Further research should address these limitations by including larger sample sizes, control groups, and longer follow-up periods. Studies combining orthoses with various conservative therapies would provide more comprehensive insights into their synergistic effects. Additionally, more objective measures of activity and lameness, such as gait analysis or motion capture technology, should be incorporated to minimize subjectivity.

## 5. Conclusions

Based on the results, it appears that knee orthoses may support better rehabilitation outcomes in dogs with CCL ruptures. The four-scale method proved to be an effective and accessible tool for evaluating weight distribution across limbs in dogs with a CCL rupture using orthoses. This method enabled the precise monitoring of rehabilitation progress, offering valuable insights into limb functionality and the impact of custom-made orthoses.

This study included two dogs with similar physical characteristics, both showing significant progress after 32 days of orthosis use. While the results are promising, further studies involving a larger sample of dogs are essential to validate the method and optimize its application for enhanced rehabilitation outcomes.

## Figures and Tables

**Figure 1 animals-15-00545-f001:**
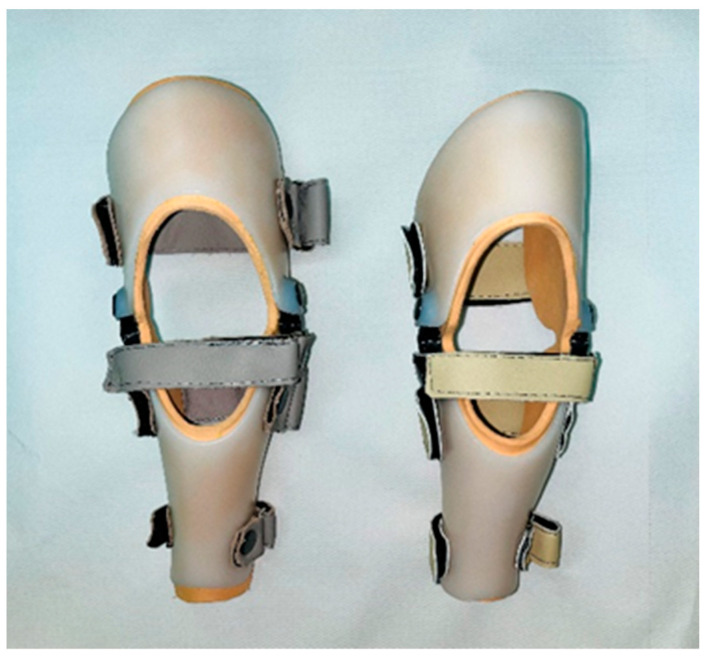
Custom-made knee orthosis for dogs.

**Figure 2 animals-15-00545-f002:**
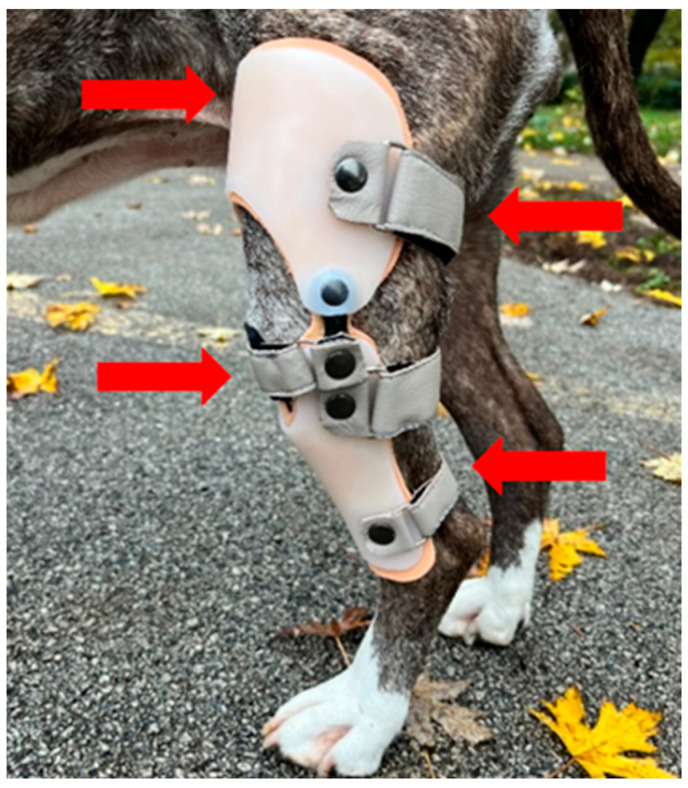
Four-point force orthosis for anteroposterior stabilization. Red arrows show the direction of force action.

**Figure 3 animals-15-00545-f003:**
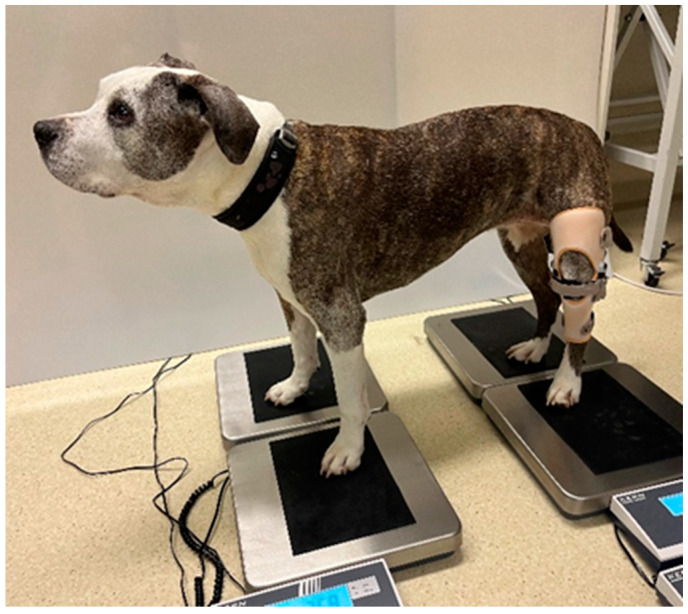
Four-scale method measurement with orthosis.

**Figure 4 animals-15-00545-f004:**
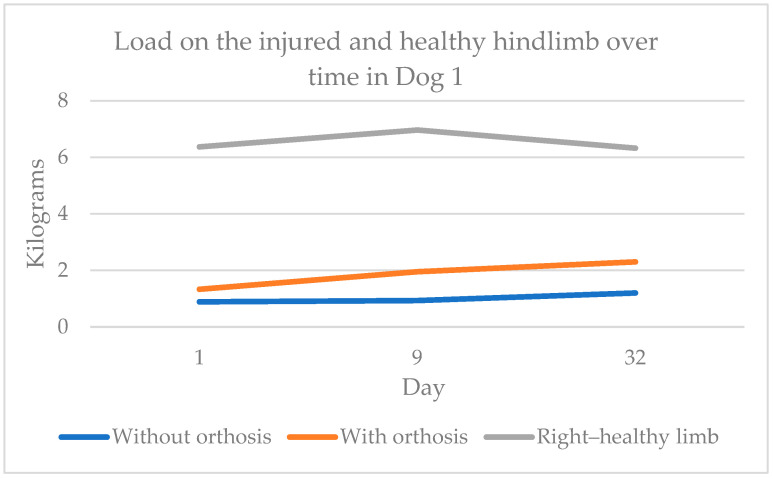
Weight transfer to the injured and healthy hindlimbs and with and without orthosis over time in Dog 1.

**Figure 5 animals-15-00545-f005:**
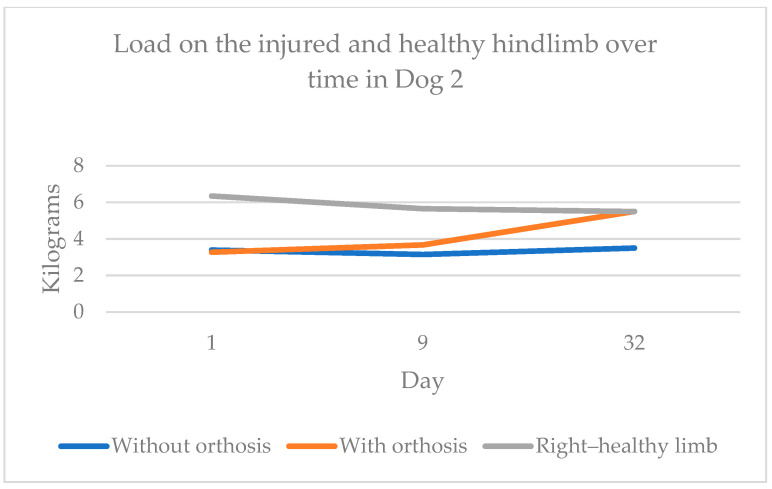
Weight transfer to the injured and healthy hindlimbs with and without orthosis over time in Dog 2.

**Figure 6 animals-15-00545-f006:**
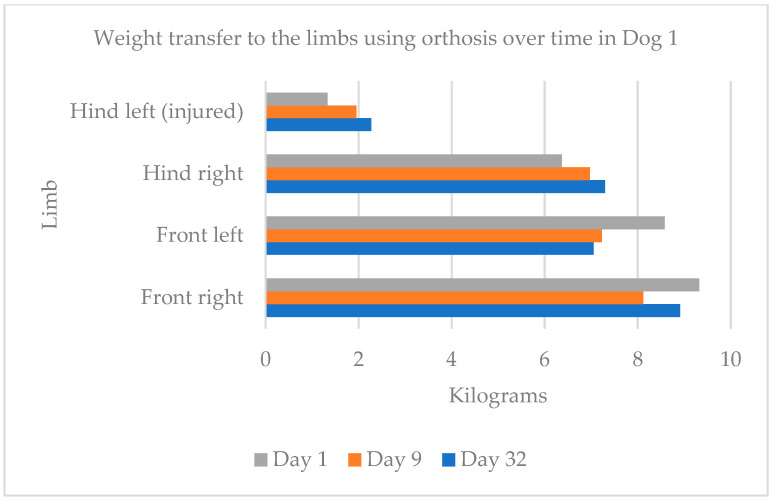
Weight transfer to the limbs using the orthosis over time in Dog 1.

**Figure 7 animals-15-00545-f007:**
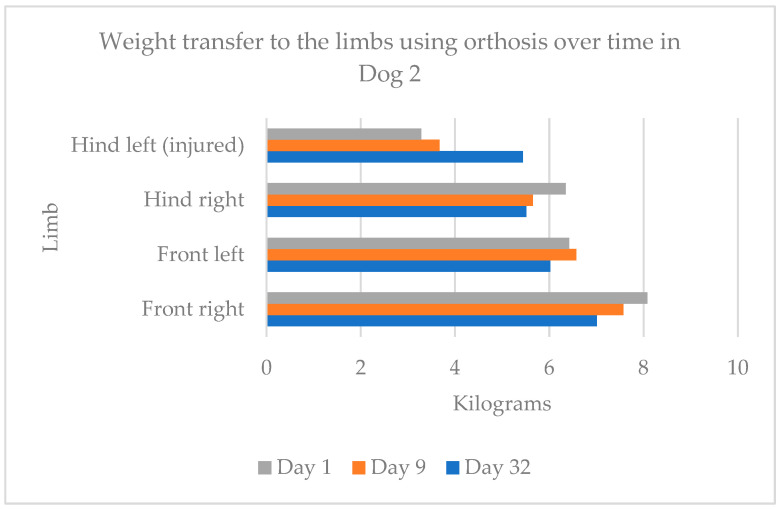
Weight transfer to the limbs using the orthosis over time in Dog 2.

**Table 1 animals-15-00545-t001:** Data of the dogs included in the study.

Physical Characteristics	Dog 1	Dog 2
Breed	Mixed breed	American Staffordshire Terrier
Age (years)	10	13
Height (top of shoulder blade) (cm)	57	48
Weight (kg)	25.62	23.89
Injured limb	Left hind	Left hind
Injury	Cranial cruciate ligament rupture	Cranial cruciate ligament rupture
Date of injury	20 September 2024 (one month before receiving the orthosis)	16 August 2024 (two months before receiving the orthosis)
Medication	Antinol (once a day), Flexadin forte (once a day)	Antinol (once a day), Palmitoylethanolamide and quercetin (Alevica) 2.5 mL (once a day), CBD 5% 4 drops (twice a day)
Other conservative treatment methods	/	Therapeutic massage (twice a month), laser therapy (once a week)
Other acute and chronic diseases	/	/
Received mobility aid	Knee orthosis	Knee orthosis
Weight of orthosis (g)	131	136

**Table 2 animals-15-00545-t002:** Questionnaire results for Dog 1 and Dog 2.

Survey Questions	Dog 1			Dog 2		
Day	Day 1	Day 9	Day 32	Day 1	Day 9	Day 32
Limping assessment—Lameness	4	3	2	3	2	2
Limping assessment—Pain of palpation	4	1	1	3	2	1
Limping assessment—Weight bearing	3	3	2	3	3	2
Average walked distance (km/day)	1	2	3,5	1	2	3
Average number of walks (per day)	2	2	2	2	2	2
Average duration of a walk (min)	30	45	60	30	30	45
Walking style	Slow	Slow	Fast	Slow	Slow	Fast
Movement assessment	Poor	Good	Good	Moderate	Good	Good
General activity level	Medium	Medium	Medium	Medium	Medium	Medium

## Data Availability

The dataset is available on request from the authors.

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
