# Peer review of "The Impact of Knee Orthoses on Lameness and Weight Distribution in Canine After Rupture of the Cranial Cruciate Ligament"

_animals, 2025, doi:10.3390/ani15040545_

Round 1

Reviewer 1 Report

Comments and Suggestions for Authors

This paper describes two clinical cases in which orthoses were chosen as the treatment instead of surgery, which is the conventional approach for cranial cruciate ligament rupture in dogs.

It is worth emphasizing that this is a highly interesting topic to focus future research efforts on. The treatment of cranial cruciate ligament rupture without surgery, the most common orthopedic condition in dogs, is a field that remains underexplored. Developing alternative treatment options for patients unable to undergo surgery due to pre-existing conditions or financial constraints could provide significant value. This makes the reported cases particularly relevant and noteworthy.

The case report is methodologically well-designed, the data are appropriate, and the results are clearly presented. However, a few minor corrections are needed:

  1. Line 60: Correct the typographical error in "simi-lar."
  2. Line 66: Correct the typographical error in "control-ling."
  3. In the bibliography, citation number 11 ” DVM, E.B. Canine Orthopedic Devices. Today’s Veterinary Practice 2016” : Please verify its accuracy, as it may not be correct.

From this reviewer’s perspective, the methodology and overall structure of the case report are well-executed and appropriate.

Additional Feedback:

  1. What is the main question addressed by the research?

This case report evaluates the effectiveness of orthoses as an alternative to surgery for the treatment of cranial cruciate ligament rupture in dogs.

  1. Is the topic original or relevant in the field, and if so, why?

This topic is highly relevant and underexplored, as noted by this reviewer. A significant number of patients cannot undergo surgery due to pre-existing health conditions that pose elevated risks, financial limitations of the owners, or other factors. It is essential to ensure that these animals also have access to effective and high-quality treatments.

  1. What does it add to the subject area compared to other published material?

To the best of this reviewer’s knowledge, there is very limited research in this area, making this one of the few prospective studies available. As such, it provides valuable insights into a topic with significant clinical relevance.

  1. Are the conclusions consistent with the evidence and arguments presented, and do they address the main question posed?

Yes, the conclusions are consistent with the evidence provided and appropriately address the main objectives of the study.

  1. Are the references appropriate?

Yes, the references are relevant and suitable for this paper, except for citation number 11, which should be reviewed for accuracy.

Author Response

Dear reviewer,

we would like to thank you for your comments. We have taken the comments into account.

Line 60: Correct the typographical error in "simi-lar."

Authors: Corrected to similar.

Line 66: Correct the typographical error in "control-ling."

Authors: Corrected to controling.

In the bibliography:  citation number 11 ” DVM, E.B. Canine Orthopedic Devices. Today’s Veterinary Practice 2016” : Please verify its accuracy, as it may not be correct.

Authors: Thank you for pointing this out. We corrected the reference 11 to

Carr, B.J.; Dycus, D.L. Canine Orthopedic Devices. Todays. Vet. Pract. 2016, 6, 117-125.

Thank you very much for your comments and feedback.

Reviewer 2 Report

Comments and Suggestions for Authors

Major Comments: 

Simple Summary: 

  • Lines 20-22: Recommend changing to: “.. knee orthoses have the potential to be an effective... and may improve the quality of life..” ; given this is a case report without control dogs nor blinding, the current claim is inappropriate.  

Abstract: 

  • Lines 33-40: The authors should clarify if these changes in weight-bearing and symmetry occurred with and/or without the orthoses in place.  

Introduction: 

  • Lines 64-76: Additional references should ideally be included, several have evaluated canine stifle orthoses for complications and efficacy (see some below): 

  • Rosen S et al, Prospective evaluation of complications associated with orthosis and prosthesis use in canine patients, Frontiers in Veterinary Science 2022.  

  • Torres BT et al, Pelvic limb kinematics in the dog with and without stifle orthosis, Veterinary Surgery, 2017.  

  • Bertocci GE et al, Biomechanics of an orthosis-managed CCL-deficient stifle joint predicted by use of a computer model, AJVR 2017. 

  • Murakami S et al, Alterations in the GRF of dogs during trot after immobilization of the stifle joint: an experimental study, J Vet Med Sci 2021. 

  • Hart JL et al, Comparison of owner satisfaction between stifle joint orthoses and tibial plateau leveling osteotomy for the management of CCL disease in dogs, JAVMA 2016.  

  • Lines 73-75: No reference provided for this claim, I am unaware of any literature to support this claim in working/sporting dogs.  

  • Lines 79-82: Given this is a case report without a control group nor blinding, recommend adjusting the aim, eg: “to evaluate the potential effectiveness.. 

Materials & Methods: 

  • The authors should provide any instructions provided to the owners regarding orthotic use, for instance: did they gradually increase the time spent in the orthotic, how many hours per day did the dogs where the orthoses, etc? 

Discussion: 

  • Lines 196-213: The authors should add to discussion/limitations: lack of blinding, any additional therapy variations (eg: when were the joint supplements initiated, some of these take 2-3 months for full effectiveness). 

Conclusions:  

  • Lines 215-216: The authors should clarify that dogs only increased weight-bearing when the orthotic was in place and change claim to “may support”.

Author Response

Dear Reviewer!

Thank you for your comments. We certainly believe that the comments were meaningful and have taken them into account.

Lines 20-22: Recommend changing to: “.. knee orthoses have the potential to be an effective... and may improve the quality of life..” ; given this is a case report without control dogs nor blinding, the current claim is inappropriate. 

Authors: We changed accordingly in Lines 20-22: The results show that knee orthoses have the potential to be an effective non-surgical alternative for the treatment of ligament injuries and may improve the quality of life of dogs.

Lines 33-40: The authors should clarify if these changes in weight-bearing and symmetry occurred with and/or without the orthoses in place. 

Authors: We have added your recommendations to the Abstract and wrote: »in both dogs when they wore orthoses« in Line 33.

Lines 64-76: Additional references should ideally be included, several have evaluated canine stifle orthoses for complications and efficacy (see some below):...

Authors: Line 81-83: Three of the suggested references have been added (12-14)

Rosen S et al, Prospective assessment of complications associated with the use of orthoses and prostheses in dogs, Frontiers in Veterinary Science 2022.;

Torres BT et al, Kinematics of the pelvic limb in dogs with and without knee orthoses, Veterinary Surgery, 2017.;

Hart JL et al, Comparison of owner satisfaction between stifle joint orthoses and tibial plateau leveling osteotomy for the management of CCL disease in dogs, JAVMA 2016.), especially regarding the complications that can be claimed.

The findings of the study Bertocci GE et al, Biomechanics of an orthosis-managed CCL-deficient stifle joint predicted by use of a computer model, AJVR 2017, are already included in the discussion (line: 214-216).

The study Murakami S et al, Changes in GRF of dogs during trotting after knee joint immobilization: an experimental study, J Vet Med Sci 2021, does not seem to make sense for this article, as it describes a different method (Force plate). It can be used for the next study in development, in which we measure the load on the prosthesis in an amputated dog with a force plate.

Lines 73-75: No reference provided for this claim, I am unaware of any literature to support this claim in working/sporting dogs.

Lines 79-82: Given this is a case report without a control group nor blinding, recommend adjusting the aim, eg: “to evaluate the potential effectiveness..

Authors: We also considered the comments in lines 73-75, so we deleted  », particularly for active animals or those participating in sport activity« and for comments in lines 79-82 we added the word »potential« in line 85.

Comment: The authors should provide any instructions provided to the owners regarding orthotic use, for instance: did they gradually increase the time spent in the orthotic, how many hours per day did the dogs where the orthoses, etc?

Authors: In Materials & Methods, we added instructions for owners, written in lines 124-130:

On receipt of the orthosis, owners were provided with instructions on the correct fitting of the orthosis, cleaning, and acclimating their pet to the orthosis. They were advised to use the orthosis only during walks and in a controlled environment for the first few days. Walks were to be adjusted based on the animal's condition. During the first week, the orthosis was recommended for 10-15 minutes per walk, with the duration gradually increasing by 10 minutes if no complications arose. Additionally, owners were instructed to examine the limbs before and after applying the orthosis to check for any skin lesions.

Comment: Lines 196-213: The authors should add to discussion/limitations: lack of blinding, any additional therapy variations (eg: when were the joint supplements initiated, some of these take 2-3 months for full effectiveness).

Authors: In the discussion, we added the time frame for the start and duration of conservative methods and the use of oral supplements. Lines 220 – 229: »This may be attributed to the additional conservative therapies (therapeutic massage and laser therapy) that Dog 2 began receiving one week after the injury and throughout the time between measurements, while Dog 1 did not. In the study by Wucherer et al., (2013) [20], they found that conservative methods of treating a CCL rupture (physical therapy, weight loss and NSAID administration) improved the health of the animals, but surgical methods achieved better results.

Both dogs also received oral joint-strengthening supplements, such as Antinol, after their injury. This suggests that combining orthotic use with complementary rehabilitation methods and oral supplements may enhance recovery outcomes.

Comment: Lines 215-216: The authors should clarify that dogs only increased weight-bearing when the orthotic was in place and change claim to “may support”.

Authors: In Conclusions, we added the results, »it appears that knee orthoses may support better rehabilitation” (line 244).

Thank you very much for your comments and feedback, which improved the article.

Reviewer 3 Report

Comments and Suggestions for Authors

A brief summary: This study aims to evaluate the effectiveness of knee orthoses in the rehabilitation of recent cranial cruciate ligament (CCL) rupture in dogs, focusing on their impact on body weight distribution across limbs over a one-month period. Utilizing a combination of the four-scales method and owner questionnaires, the paper highlights the growing use of orthoses in veterinary medicine for stabilizing joints, relieving pain, and promoting functional recovery. Its main contributions include providing insights into the therapeutic benefits of orthoses as a non-surgical option and their role in improving biomechanics and facilitating recovery, making it a valuable option for surgically-restricted patients.

General concept comments
Article: One area of weakness of the manuscript is a low number of animals, as well as a short follow-up time, but this has been adressed by the authors. If possible, a longer-term time point should be added. In general, recovery from surgical procedures used to treat CCL in dogs takes approximately 3-4 months, so this period would be more appropriate to evaluate the effectiveness and treatment outcomes of orthoses. Another issue which could be adressed is the longevity of injury.

No data on the time of injury was given rather than 'recent'. Is the data for injury longevity othere than recent availabe? Is the data for meniscal injury available or were meniscal click signs absent in both dogs? It's also well-reported that dogs treated with conservative means other than orthoses have improved gait over time (Budsberg SCLong term temporal evaluation of ground reaction forces during development of experimentally induced osteoarthritis in dogsAm J Vet Res200162:12071211.; Wucherer KL, Conzemius MG, Evans R, Wilke VL. Short-term and long-term outcomes for overweight dogs with cranial cruciate ligament rupture treated surgically or nonsurgically. J Am Vet Med Assoc. 2013 May 15;242(10):1364-72. doi: 10.2460/javma.242.10.1364. PMID: 23634680.).

Review: It's well-published that about 30% (data varies) of dogs develop a meniscal injury prior to surgery, but as a consequence of CCLR. About 5% of dogs with intact menisci develop a post-surgical meniscal rupture. Is there any data in the literature adressing this phenomenon in regards to orthoses use or is this a blind spot? Was an orthopaedic exam done at the last time-point to screen for possible meniscal injury (meniscal click)?

Did the dogs have rotational instability at the time of presentation and was a pivot-shift detected in the study dogs? The article mentions the benefits of orthotic stabilization of joints and data on stifle laxity in the test dogs and rotational stability of the affected joints would be a valuable addition to the manuscript (Lampart M, Park BH, Husi B, Evans R, Pozzi A. Evaluation of the accuracy and intra- and interobserver reliability of three manual laxity tests for canine cranial cruciate ligament rupture-An ex vivo kinetic and kinematic study. Vet Surg. 2023 Jul;52(5):704-715. doi: 10.1111/vsu.13957. Epub 2023 May 5. PMID: 37144831.)

Specific comments None.

The manuscript is clear, well-written, relevant for the field and presented in a well-structured manner. 

The cited references are mostly recent publications (within the last 5 years) and relevant. It includes one (1) self-citation, which is relevant to the paper and appropriate.

Is the manuscript scientifically sound and is the experimental design appropriate to test the hypothesis? It would be appropriate to compare dogs undergoing surgical treatment with dogs undergoing conservative treatment with orthoses and dogs undergoing conservative trreatment without orthoses.

The manuscript’s are results reproducible based on the details given in the methods section.

Are the figures/tables/images/schemes are appropriate and easy to understand.

The conclusions are consistent with the evidence and arguments presented.

Author Response

Dear reviewer!

Thank you for your comments, which have improved the article.

We fully acknowledge the small sample size in our study; however, at the time of research, we were unable to recruit more animals with similar injuries. This limitation has been addressed in the manuscript. For future research, we are planning a similar study with a larger sample size, including animals that have undergone surgery.

The time of injury for each dog is recorded in Table 1 (Date of injury), therefore, we have avoided repeating this information in the text.

Before the application of orthoses, the dogs were examined by veterinary orthopaedics at private clinics in Slovenia (Ljubljana and Postojna). X-rays and manual laxity tests were performed (as noted in lines 94-100). No other injuries or pathologies were identified, apart from cranial cruciate ligament (CCL) rupture. Due to the animals' age and the potential risks associated with anaesthesia, laparoscopic meniscal examination was not performed. After the orthoses were applied, the dogs underwent a veterinary evaluation to ensure proper fit and a certificate of orthotic suitability was issued. However, a veterinary examination to assess potential meniscal damage was not performed. While owners were advised to seek further diagnostic evaluation, the final decision remained at their discretion. We add the following text: »The dogs underwent X-rays and a manual laxity test performed by veterinarians from two private veterinary clinics to confirm the diagnosis.«

Comment: It's also well-reported that dogs treated with conservative means other than orthoses have improved gait over time.

Authors: We agree with your comment and we have added to the discussion (lines: 221-224) the study by Wucherer KL et al., 2013. We added the following: »In the study by Wucherer et al., (2013) [20], they found that conservative methods of treating a CCL rupture (physical therapy, weight loss and NSAID administration) improved the health of the animals, but surgical methods achieved better results.«

Thank you for your detailed observations.

Reviewer 4 Report

Comments and Suggestions for Authors

This case report discusses two cases of cranial cruciate ligament rupture in dogs treated with stifle orthoses. The evaluation was made by owner questionnaires and measurement of the weight distribution using a four-scale method.

The topic of cruciate ligament rupture and orthoses in dogs is still relevant, but the statement that there was an improvement in weight transfer - in two dogs where only one has really improved - is a little too optimistic.

Otherwise, the manuscript is clear and presented in a well-structured manner, the cited references are ok, and the case report is scientifically sound.

The results are mainly reproducible based on the details given in the methods section.

There is one small comment to the authors:

“Line 129 or Figure 3.:

Can you please describe the head posture more detailed, because it surely makes a difference in weight distribution between front and hindlimbs if the head is slightly more up or down. A picture taken at the time point of reading the scales would be perfect to ensure a completely correct measurement for further studies. Maybe also mention this in the discussion (Correct stance – looking in front, head at the same height)”

The figures, tables and images are appropriate and show the data in a clear way. They are quite easy to interpret and understand.

The discussion and the conclusion are consistent with the evidence and arguments presented, even though I would emphasize to discuss the head posture (also the height) in the discussion – as mentioned above.

In conclusion there is an overall benefit to publish this case report because the use of orthoses is still discussed in the field of orthopedics in dogs and this case reports encourages its use, even though there still is a need of further studies with bigger sample sizes.

Author Response

Dear Reviewer!

Thank you for the comments that have improved the article.

Comment: The topic of cruciate ligament rupture and orthoses in dogs is still relevant, but the statement that there was an improvement in weight transfer - in two dogs where only one has really improved - is a little too optimistic.

Authors: We agree with this comment, and this limitation has been addressed in the "Limitations" section, where we specifically mention that the small sample size limits the generalizability of the results (Lines: 230-231). Thank you for your feedback.

In the future, we are planning a similar study with a larger sample size, including animals that have undergone surgery.

Comment: “Line 129 or Figure 3.:

Can you please describe the head posture more detailed, because it surely makes a difference in weight distribution between front and hindlimbs if the head is slightly more up or down. A picture taken at the time point of reading the scales would be perfect to ensure a completely correct measurement for further studies. Maybe also mention this in the discussion (Correct stance – looking in front, head at the same height)”

Authors: We have taken the comments into account and added an explanation of the head position (lines: 148-150: The dog's posture was carefully monitored to ensure all four limbs were placed correctly on the corresponding scale with the head straight forward and at the same height between each measurement (Figure 3). If the head was lower or higher or rotated to the left or right, there would be an incorrect distribution of weight on the limbs.) and the discussion (line: 209-211: Great attention was paid to the posture of the dogs during all measurements. The height and position of the head (straight forward) were carefully monitored to ensure that there was no incorrect weight shift ). Unfortunately, we do not have a photo taken at the time the results were read, but the position of Figure 3 is a very good approximation of the position the dogs had.

Thank you very much for your comments and feedback.

Round 2

Reviewer 2 Report

Comments and Suggestions for Authors

Thank you for providing these revisions, you have addressed the previous concerns and strengthened the clarity of the manuscript.